# A Hydrophobic Ratiometric Fluorescent Indicator Film Using Electrospinning for Visual Monitoring of Meat Freshness

**DOI:** 10.3390/foods14132200

**Published:** 2025-06-23

**Authors:** Xiaodong Zhai, Xingdan Ma, Yue Sun, Yuhong Xue, Wanwan Ban, Wenjun Song, Tingting Shen, Zhihua Li, Xiaowei Huang, Qing Sun, Kunlong Wu, Zhilong Chen, Wenwu Zou, Biao Liu, Liang Zhang, Jiaji Zhu

**Affiliations:** 1Agricultural Product Processing and Storage Lab, School of Food and Biological Engineering, Jiangsu University, Zhenjiang 212013, China; zhai_xiaodong@ujs.edu.cn (X.Z.); xingdanma@163.com (X.M.); 1000006728@ujs.edu.cn (Y.S.); xue_yuhong@163.com (Y.X.); 13275356807@163.com (W.B.); 18952992762@163.com (W.S.); shentingtingstt@ujs.edu.cn (T.S.); lizh@ujs.edu.cn (Z.L.); 2Wencheng Institution of Modern Agriculture and Healthcare Industry, Wencheng, Wenzhou 325300, China; kunlongwu@126.com (K.W.); zhilongchen@sina.com (Z.C.); 13968830330@163.com (W.Z.); liubiao17@163.com (B.L.); liangzhang_xj@163.com (L.Z.); 3School of Electrical Engineering, Yancheng Institute of Technology, Yancheng 224051, China; zhujiaji@ycit.edu.cn

**Keywords:** hydrophobic, ratiometric fluorescent film, electrospinning, food freshness, intelligent packaging

## Abstract

A ratiometric fluorescent film with high gas sensitivity and stability was developed using electrospinning technology for monitoring food spoilage. 5(6)-Carboxyfluorescein (5(6)-FAM) was used as the indicator, combined with the internal reference Rhodamine B (RHB), to establish a composite ratiometric fluorescent probe (FAM@RHB). The hydrophobic fluorescent films were fabricated by incorporating FAM@RHB probes into polyvinylidene fluoride (PVDF) at varying molar ratios through electrospinning. The FR-2 film with a 2:8 ratio of 5(6)-FAM to RHB exhibited the best performance, demonstrating excellent hydrophobicity with a water contact angle (*WCA*) of 113.45° and good color stability, with a Δ*E* value of 2.05 after 14 days of storage at 4 °C. Gas sensitivity tests indicated that FR-2 exhibited a limit of detection (*LOD*) of 0.54 μM for trimethylamine (TMA). In the application of monitoring the freshness of pork and beef at 4 °C, the fluorescence color of the FR-2 film significantly changed from orange–yellow to green, enabling the visual monitoring of meat freshness. Hence, this study provides a new approach for intelligent food packaging.

## 1. Introduction

Meat is a crucial nutritional source, rich in proteins, fats, and minerals for human nutrition [1]. However, it easily undergoes biochemical reactions like microbial contamination, enzymatic reactions, and lipid oxidation, which compromise its freshness and lead to spoilage [2,3,4]. Total volatile basic nitrogen (*TVB-N*) is a crucial indicator of meat spoilage, mainly composed of volatile substances like ammonia (NH_3_), dimethylamine (DMA), and trimethylamine (TMA) [2,5,6]. Smart packaging can conduct real-time monitoring and feedback on the volatile compounds generated during the deterioration of food. It can dynamically indicate the freshness of food in an intuitive way, such as through color changes, making it convenient for both producers and consumers to use [7,8,9,10,11,12].

Freshness indicators are functional labels that intuitively reflect food freshness through signal changes like color and electrochemical indicators, including gas or pH-sensitive indicators [13], microbial indicators [14], time–temperature indicators [15], and so on. Colorimetric indicators are more convenient to use compared to electrochemical indicators because they can directly indicate freshness through color changes. Their responsive properties are harnessed to monitor spoilage: for example, decomposing meats and fish release volatile amines (e.g., trimethylamine and putrescine) that trigger distinct chromatic or fluorescent shifts of colorimetric indicators. These visual markers provide intuitive, real-time feedback on amine concentration changes, enabling rapid freshness evaluation without complex instrumentation.

Fluorescence sensors, known for their simplicity, cost-effectiveness, and visual output, are increasingly being integrated into smart packaging solutions [16,17,18,19]. However, traditional monochromatic fluorescent probes, which operate on a simple “off-on” mechanism, suffer from limitations such as low sensitivity, poor visual differentiation, and practical portability issues [4,20,21,22]. These probes are also vulnerable to environmental and instrumental interference, and the subtle changes in single-color brightness are often imperceptible to the human eye [4,16,23]. Additionally, fluorescent indicators are prone to aggregation quenching, and the indicator films suffer from low self-stability in the packaging environment of high-humidity meat products. These issues significantly limit the practical prospects of these fluorescent indicator films.

In contrast, ratiometric fluorescent sensors that utilize two luminophores (one indicator and one internal reference) possess superior self-calibration capabilities and exhibit enhanced sensitivity and accuracy [24,25,26,27]. Many studies have confirmed that ratiometric fluorescent probes can provide built-in self-calibration by calculating the intensity ratio of two fluorescent signals, thereby enabling more accurate quantification [28,29,30,31,32].

Rhodamine B (RHB), a cost-effective fluorescent dye, exhibits red fluorescence with a high quantum yield, strong absorption coefficient, pH-resistant fluorescence, and stable chemical properties, making it an ideal self-reference signal [33,34]. Some scholars used Rhodamine B as an internal reference, salicylamide as a pH indicator, and polyvinyl alcohol as the base material to prepare a pH-responsive intelligent packaging system for the real-time monitoring of the freshness of seafood [35]. Some scholars used Rhodamine B (internal reference) and fluorescein isothiocyanate (indicator) to detect histamine in raw meat, a combination that exhibited a notable color transition from yellow to green [36]. In addition, 5(6)-Carboxyfluorescein (5(6)-FAM) is a fluorescent indicator with high sensitivity and stability, featuring favorable optical properties, ease of biomolecule conjugation, and low cytotoxicity, which make it ideal for biomedical applications [37].

Electrospinning technology has been widely used in the preparation of polymer fiber films. The intelligent indicator films developed through electrospinning possess advantages such as high encapsulation efficiency, adjustable porosity, easy manipulation of chemical compositions, and flexible material selection. The porous network of electrospun nanofibers increases the number of active sites for interaction with target molecules, thus improving the sensitivity of the films [38,39]. Meanwhile, the semi-embedded hydrophobic matrix prepared by electrospinning technology can enhance the stability of the indicator films in meat packaging. However, there are few studies that combine fluorescent sensors with electrospinning to develop intelligent packaging for monitoring food freshness [40,41,42,43].

Our recent study successfully developed a ratiometric fluorescent sensor using as-synthesized carbon dots as the probes [44]. In comparison, this study employed both Rhodamine B and 5(6)-FAM as probes due to their commercial availability, making s the fluorescent sensor more conducive to commercialization.

This study involved the development of a ratiometric fluorescent probe, FAM@RHB, by combining RHB as the internal reference and 5(6)-FAM as the indicator. The optimal ratio of these two components was systematically determined. Subsequently, FAM@RHB was incorporated into PVDF nanofibers using electrospinning technology to develop and characterize a hydrophobic ratiometric fluorescent film. The sensitivity of the indicator film to trimethylamine and its stability were the primary focus of this study. Finally, it was applied for monitoring the freshness of pork and beef.

## 2. Materials and Methods

### 2.1. Materials

5(6)-Carboxyfluorescein (5(6)-FAM, purity: 95%), Rhodamine B (RHB, purity: 99%), and ammonia solution (NH_3_) (25–28 wt%) were obtained from Shanghai Macklin Biochemical Technology Co., Ltd. (Shanghai, China). Polyvinylidene fluoride (PVDF, purity: ≥99.9%) was supplied from Shanghai Yuanye Bio-Technology Co., Ltd. (Shanghai, China). N,N-Dimethylformamide (DMF, purity: ≥99.5%) and magnesium oxide (MgO, purity: 99%) were purchased from Aladdin Biochemical Technology Co., Ltd. (Shanghai, China). Acetone (purity: ≥99.5%), boric acid (H_3_BO_3_, purity: ≥99.5%), and hydrochloric acid (HCl, 36%~38%) were acquired from Sinopharm Chemical Reagent Co., Ltd. (Shanghai, China). Deionized water was used throughout the experiments. All chemicals were of analytical grade and used as received without further purification.

### 2.2. Preparation of Ratiometric Fluorescent Probes

Initially, 10 mg of 5(6)-FAM was dissolved in 10 mL of deionized water under continuous magnetic stirring at room temperature for 10 min and then ultrasonicated for 5 min, yielding a homogeneous yellow–green solution. Simultaneously, 10 mg of RHB was dissolved in 10 mL of absolute ethanol through magnetic stirring and then ultrasonicated for 10 min, resulting in a uniform orange solution.

The process of optimizing the ratio of 5(6)-FAM to RHB was as follows: firstly, the 5(6)-FAM and RHB solutions were mixed at varying volume ratios (1:9, 2:8, 3:7, 4:6, and 5:5) for the preparation of ratiometric fluorescent probes. A 10 μL aliquot of the ratiometric fluorescent probe was diluted with 3 mL of ultrapure water. A specific volume of TMA solution was then added to the mixture, which was thoroughly mixed and allowed to react for 3 min. The fluorescence spectra of the FAM@RHB mixtures were recorded using a fluorescence spectrophotometer both before and after the reaction. The parameters were set as follows: excitation wavelength of 490 nm, emission wavelength range of 490–650 nm, scanning speed of 1000 nm/min, excitation bandwidth and emission bandwidth of 10 nm, and gain of 7 steps [45]. By comparing the fluorescence intensity ratios of the two emission peaks before and after the reaction, the FAM@RHB ratiometric fluorescent probe with the highest degree of reaction was identified.

### 2.3. Preparation of Electrospun Ratiometric Fluorescent Films

To create the electrospinning substrate solution, 1.5 g of PVDF powder was dissolved in 10 mL of a binary solvent system consisting of DMF and acetone with a volume ratio of 4:1. The mixture was magnetically stirred at 40 °C for 3 h to obtain a transparent PVDF polymer solution. Subsequently, 2% (*w*/*v*) of the prepared FAM@RHB fluorescent probe solutions were individually incorporated into the PVDF solution [8]. Each mixture was subjected to magnetic stirring for 30 min followed by 30 min of ultrasonication to ensure complete homogenization and bubble removal, yielding clear and transparent red-colored electrospinning solutions.

The ratiometric fluorescent films were fabricated using an electrospinning apparatus. The polymer solution was loaded into a 10 mL syringe equipped with a stainless steel needle (0.91 mm outer diameter, 0.58 mm inner diameter). The electrospinning parameters were optimized as follows: a positive voltage of 15 kV, a negative voltage of −3 kV, a chamber temperature of 35 °C, a relative humidity of 45%, a needle-to-collector distance of 20 cm, a solution flow rate of 0.001 mm/s, and a receiver rotation speed of 20 r/min. To enable seamless film collection, the drum receiver was lined with silicone-coated greaseproof paper, ensuring optimal deposition of the electrospun fibers under the high-voltage electrostatic field [46]. The resulting fluorescent films were designated as FR-1, FR-2, FR-3, FR-4, and FR-5, corresponding to the respective volume ratios of 5(6)-FAM to RHB (1:9, 2:8, 3:7, 4:6, and 5:5) incorporated in each film.

### 2.4. Scanning Electron Microscopy (SEM) of Indicator Films

The samples were trimmed into appropriate sizes and mounted onto SEM stubs using a conductive adhesive, ensuring flat surface attachment. Prior to imaging, the samples were sputter-coated with a thin layer of gold for 5 min to enhance conductivity [47]. The surface morphology and fiber architecture were examined using field emission scanning electron microscopy (S-4800, Hitachi High Technologies Corporation, Tokyo, Japan) at an accelerating voltage of 5 kV. Fiber diameter distribution was quantitatively analyzed using Nano Measurer 1.2 software.

### 2.5. Laser Confocal Image of the Indicator Film

For confocal microscopy analysis, the indicator films were precisely sectioned into thin strips and mounted onto glass slides. The specimens were meticulously placed at the central area of the slide and then sealed with a coverslip to guarantee consistent flatness across the sample surface [48]. The edges were secured with laboratory tape to prevent movement during imaging. The prepared slides were then inverted and positioned on the microscope stage for imaging. Fluorescence imaging was performed using a laser scanning confocal microscope (TCS SP5, Leica, Germany) with an excitation wavelength of 488 nm.

### 2.6. Fourier Transform Infrared Spectroscopy (FTIR) of Indicator Films

The chemical composition of the indicator films was characterized using FTIR in attenuated total reflectance (ATR) mode [49]. Spectra were acquired over a wavenumber range of 525–4000 cm^−1^ with a spectral resolution of 4 cm^−1^. Each measurement was performed with 32 scans at room temperature to ensure an optimal signal-to-noise ratio. Prior to analysis, the ATR crystal was carefully cleaned with ethanol, and background spectra were collected. The spectra that were obtained were processed with baseline correction and normalization using the instrument-specific software (omnic 8.2), facilitating comparative analysis.

### 2.7. Water Contact Angle (WCA) of the Indicator Film

Static water contact angle measurements were conducted to assess the surface hydrophobicity of the indicator films [50]. Film samples were precisely cut into 20 × 20 mm squares and mounted onto glass slides using double-sided adhesive tape, ensuring a flat and wrinkle-free surface. The prepared samples were placed on a leveled stage of the contact angle goniometer. A 5 μL droplet of deionized water was carefully dispensed onto the film surface using a precision microsyringe. The droplet profile was captured using a high-speed camera equipped with image analysis software. The WCA was determined using the Young–Laplace fitting method, with measurements taken at three different locations on each sample to ensure reproducibility. All measurements were conducted under ambient conditions (25 ± 1 °C, 50 ± 5% relative humidity).

### 2.8. Stability of Indicator Films

The color stability of the indicator films was systematically evaluated through controlled storage experiments. Film samples were stored in a temperature-controlled chamber maintained at 4 ± 0.5 °C for a duration of 14 days. Colorimetric analysis was performed at two-day intervals to monitor any chromatic changes. For fluorescence imaging, the films were placed in a UV chamber equipped with 365 nm excitation and photographed using a digital camera with fixed parameters. The camera position and imaging conditions were rigorously maintained throughout the experiment to ensure consistency and minimize experimental variability. The *L** (lightness), *a** (red-to-green), and *b** (blue-to-yellow) values of the photos were using a calibrated imaging system [51]. Triplicate measurements were performed for each sample, and the mean values were calculated to ensure statistical reliability. The total color difference (Δ*E*) between initial and stored samples was calculated using the following equation:(1)$$\Delta E=\sqrt{{\left(\Delta L*\right)}^{2}+{\left(\Delta a*\right)}^{2}+{\left(\Delta b*\right)}^{2}}$$
where Δ*L** = *L** − *L*_0_*, Δ*a* = *a** − *a*_0_*, Δ*b* = *b** − *b*_0_*. *L**, *a**, and *b** are the color parameters of the indicator film after a certain storage time. *L*_0_*, *a*_0_*, and *b*_0_* are the initial color parameters of the indicator film.

### 2.9. Response of Indicator Films to TMA

TMA, a well-established biomarker for meat spoilage, was used to evaluate the gas-sensing performances of the developed indicator films [52]. A 1 × 1 cm film sample was placed in a sealed test chamber maintained under controlled conditions (25 °C, 60% relative humidity). A microliter syringe was used to inject TMA solution into the chamber, and the system was allowed to stand for 60 min to ensure a complete reaction. The concentrations of TMA in the testing chamber were 0, 2.86, 5.71, 8.57, 14.28, 20, 28.57, 57.14, 85.71, and 142.86 μM, respectively. Triplicate parallel experiments were carried out to ensure reproducibility, and the fluorescence color parameters of the indicator films were collected. The Δ*E* values were calculated according to Equation 2.1, and the average value was taken as the final result. The following is the formula used for calculating the concentration of TMA in the testing chamber:(2)$$C=\frac{W_{S}\;V_{S}\;\rho_{S}}{M\;V}$$
where *C* (mol/L) is the concentration of the analyte in the test chamber; *ρ_S_* is the density of the analyte (g/mL); *V_S_* is the volume of the analyte (mL); *Ws* is the mass fraction of the analyte; *M* is the molar mass of the analyte (g/mol); *V* is the volume of the reaction vessel (L).

The limit of detection (*LOD*) of the indicator film can be calculated by using the following formula [53]:(3)$$\mathrm{LOD}=\frac{3\sigma }{N}$$
where *σ* is the standard deviation of 13 replicate blank measurements, and *N* is the absolute value of the slope of the calibration curve.

### 2.10. Application of Indicator Films for Livestock Meat Freshness Monitoring

For the same batch of purchased meat, one part was used for application testing, while the other part was utilized to determine total volatile basic nitrogen (*TVB-N*). Both beef (loin) and pork (loin) were sampled from the longissimus dorsi muscle. Visible fat and connective tissues were excised, and sterile stainless steel molds were employed to cut the meat into 60 ± 1 g cuboids, ensuring uniform muscle fiber orientation.

FR-2 fluorescent indicator films (1.5 × 1.5 cm) were affixed to the interior top surface of each container. The thickness of the applied FR-2 film was approximately 126 ± 2 μm. The packaged samples were stored in a temperature-controlled refrigerator at 4 °C for 7 days. The fluorescence response was monitored daily using a UV chamber (365 nm excitation), with digital images captured at 24 h intervals using a standardized camera setup. To assess the effectiveness of the indicator film in evaluating meat freshness, the *TVB-N* values of pork and beef were measured. The measurements were carried out according to the Chinese National Standard GB 5009.228-2016 [54] and previous research methods [55]. To ensure that the two groups of samples were as identical as possible, we used large cuts of pork and beef from the same source. On the first day, the large cuts of pork and beef were sliced into smaller pieces of a uniform size. Portions of the pork and beef pieces were placed into packaging boxes equipped with fluorescent indicator films, while the remaining portions were placed into identical packaging boxes without indicator films. Both groups were stored under the same conditions. Samples for *TVB-N* testing were taken from the packaging boxes without indicator films.

### 2.11. Statistical Analysis

All experiments were repeated three times and the results were averaged over the three times. Significant differences between data were determined using Duncan’s multiple range test, defining *p* < 0.05 as significant. Graphs were plotted using IBM SPSS Statistics 25 and Origin 2019 analysis.

## 3. Results and Discussion

### 3.1. Optical Properties of 5(6)-FAM and RHB

Figure 1 illustrates the fluorescence emission spectra of 5(6)-FAM and RHB under varying excitation wavelengths. The maximum emission intensity of 5(6)-FAM was observed at an excitation wavelength of 490 nm, with a corresponding emission peak at 516 nm. Notably, the emission peak positions of 5(6)-FAM remained constant across excitation wavelengths ranging from 450 nm to 550 nm. Similarly, as shown in Figure 1B, RHB exhibited its maximum fluorescence intensity at an excitation wavelength of 550 nm, with an emission peak at 580 nm. The emission characteristics of RHB demonstrated wavelength independence within the excitation range of 500 nm to 600 nm. This spectral behavior can be attributed to the fundamental principles of fluorescence emission, where π-electrons transition from the lowest vibrational level of the first excited state to various vibrational levels of the ground state. The consistent emission peak positions, independent of excitation wavelength, confirm the potential of 5(6)-FAM and RHB as reliable components for ratiometric fluorescent probes. Given that 5(6)-FAM functions as the TMA-responsive component in the FAM@RHB ratiometric fluorescent probe system, 490 nm was identified as the optimal excitation wavelength for subsequent experiments. This selection ensures maximum sensitivity while maintaining the ratiometric characteristics of the probe system [56].

### 3.2. Construction and Ratio Optimization of FAM@RHB Fluorescent Probe

To identify the optimal FAM@RHB probe ratio demonstrating maximal sensitivity to TMA, various probe formulations were prepared with 5(6)-FAM to RHB volume ratios of 1:9, 2:8, 3:7, 4:6, and 5:5. The TMA responsiveness of these probes was evaluated using a 20 μM TMA solution, as illustrated in Figure 2. At the 1:9 ratio, the fluorescence intensity at 516 nm (*F*_516nm_) increased from 64 to 222, while the fluorescence intensity at 580 nm (*F*_580nm_) remained almost unchanged upon TMA exposure (Figure 2A), with a corresponding ratiometric fluorescence difference (*F*_516nm_/*F*_580nm_) of 0.35 before and after the reaction (Figure 2F). This limited responsiveness can be attributed to the relatively low concentration of 5(6)-FAM, the TMA-sensitive component, in the probe formulation.

The most pronounced response was observed at the 2:8 ratio, where *F*_516nm_ increased from 151 to 614 following TMA exposure. As shown in Figure 2B,F, this formulation exhibited the maximum ratiometric fluorescence difference (*F*_516nm_/*F*_580nm_) of 0.94. Interestingly, a further increase in the 5(6)-FAM proportion resulted in diminished responsiveness, as evidenced by gradually decreasing *F*_516nm_/*F*_580nm_ differences (Figure 2C–F). This phenomenon likely reflects the stoichiometric requirements of the reaction, where higher indicator concentrations necessitate proportionally greater TMA quantities to achieve equivalent response levels, consequently reducing the apparent sensitivity of the probe.

### 3.3. UV–Vis Spectra of FAM@RHB

The ultraviolet–visible (UV–Vis) absorption spectra of 5(6)-FAM and RHB are presented in Figure 3. The absorption spectrum of 5(6)-FAM can be categorized into two distinct regions: the ultraviolet (UV) region and the visible region. A significant portion of the absorption is observed below 300 nm, a characteristic feature of this fluorescein derivative. Additionally, 5(6)-FAM exhibits minor absorption in the visible region, with absorption bands observed at 452 nm and 480 nm, corresponding to the enol and keto forms of fluorescein, respectively [57]. The absorption peak at 230 nm is attributed to the π–π* electronic transition of the ester group [58]. The presence of these distinct absorption features in the visible region is particularly significant for the development of optical sensors, as it enables selective excitation of the fluorophore within the visible spectrum. As shown in Figure 3, RHB displays characteristic absorption features in both UV and visible regions. RHB exhibits a prominent absorption band in the visible region at 554 nm, which is primarily attributed to the C=N and C=O structures within its conjugated system. Additionally, RHB shows absorption peaks in the UV region at 258 nm and 352 nm mainly originating from its benzene ring structure [59]. These absorption features demonstrate RHB’s strong light-absorbing properties across a broad spectral range, making it suitable for various photochemical applications.

### 3.4. FAM@RHB Assay for TMA

During meat storage, microbial growth is the primary factor driving gradual spoilage. This process is accompanied by the generation of a range of characteristic spoilage gases, including TMA and NH_3_, which are widely recognized as key indicators for evaluating meat freshness [60]. Therefore, TMA was selected as a representative gas to evaluate the fluorescent color response capability of the FAM@RHB probe.

Figure 4A demonstrates the TMA-responsive characteristics of the FAM@RHB probe. As TMA concentration increased from 1.67 to 40 μM, *F*_516nm_ progressively increased, while *F*_580nm_ remained nearly constant. As shown in Figure 4D, with increasing TMA concentration, the FAM@RHB fluorescent probe exhibited a progressive color shift from red to orange and finally to light green under 365 nm UV irradiation, consistent with the variations in the emission peak intensity of the fluorescence spectrum. Under natural light, a distinct red-to-orange transition was evident. With increasing TMA concentration, the ratiometric fluorescence values (*F*_516nm_/*F*_580nm_) exhibited a strong linear correlation with TMA concentration (y = 0.079x + 0.062, *R*^2^ = 0.980), as shown in Figure 4B. These results confirm the probe’s excellent TMA detection capability.

### 3.5. Microstructure of the Indicator Film

Figure 5A presents the microscopic morphology of the indicator films. All films exhibited a three-dimensional network structure composed of interwoven nanofibers with rough surfaces, attributed to the polymer characteristics and electrospinning conditions. This surface roughness could enhance the gas sensitivity of films by increasing the effective contact area with TMA. Compared with the FR-3, FR-4, and FR-5 indicator films, the FR-1 and FR-2 films exhibited higher porosity and smaller fiber diameters. This phenomenon can be attributed to the higher loading of 5(6)-FAM in the FR-3, FR-4, and FR-5 films, which resulted in an increased number of positively charged hydrogen atoms. These hydrogen atoms may have established electrostatic interactions with the electronegative fluorine atoms on PVDF, consequently modifying the substrate’s physical properties. The FR-2 indicator film exhibited a smaller fiber diameter, providing a higher specific surface area of fibers for its subsequent favorable response to TMA.

### 3.6. Hydrophobicity of the Indicator Film

The hydrophobicity of the indicator films is closely related to their performance in high-humidity meat packaging applications. Films with stronger hydrophobicity demonstrate enhanced stability in high-humidity environments by preventing the leaching of indicator materials.

The WCA value of the unmodified PVDF film and the five FR indicator films are presented in Figure 6A. The WCA value of the blank PVDF film was 123.60°, which was higher than those of the five FR indicator films. This suggests that the incorporation of the indicator reduced the hydrophobicity of the films. From FR-1 to FR-5, as the RHB content decreased and the 5(6)-FAM content increased, the amount of hydrophilic moieties (for instance, hydroxyl groups) within the indicator films was reduced. Consequently, the indicator film WCA values increased from 99.63° to 121.69°, indicating a gradual enhancement in hydrophobicity. In addition, FR-1 exhibited significantly lower and different hydrophobicity (*p* < 0.05) when compared to the other four FR indicator films and the blank PVDF film (Figure 6B). A surface with a WCA value greater than 90° is generally considered hydrophobic. Overall, all five indicator films exhibited good hydrophobicity, demonstrating their potential for application in high-humidity meat packaging.

### 3.7. Laser Confocal Images of the Indicator Films

Figure 7 shows the laser confocal images of the indicator films. In the green channel, the green fluorescence of the FR-5 indicator film was obviously stronger than that of the FR-1 indicator film, while in the red channel, the red fluorescence of the FR-5 indicator film was obviously weaker than that of the FR-1 indicator film, which was due to the difference in fluorescence intensity caused by the gradual increase in the proportion of 5(6)-FAM in the indicator films from FR-1 to FR-5 and the gradual decrease in the proportion of RHB. In addition, according to the fluorescence distribution, it can be observed that the fluorescent probes were evenly loaded into the fibers, without an obvious aggregation phenomenon. In the bright-field image, it can be observed that the polymer fibers were well defined, which made the indicator film have a good porous structure for subsequent gas sensing.

### 3.8. FTIR Spectral Analysis of the Indicator Film

Figure 8 presents the FTIR spectra of the fluorescent probes RHB and 5(6)-FAM, as well as the FR-2 film and the blank PVDF film. In the FTIR spectrum of 5(6)-FAM, the broad absorption peak observed at 3434 cm^−1^ is assigned to the O-H stretching vibration. The characteristic absorption peak at 1697 cm^−1^ corresponds to the C=O stretching vibration of the carbonyl group. Additionally, two distinct bands near 1392 cm^−1^ are associated with the C-O-C stretching vibration in the ester-based structure [61]; in the IR spectrum of RHB, the peak at 2990 cm^−1^ is attributed to the C-H bending vibration. The characteristic absorption peak at 1699 cm^−1^ arises from the C=O stretching vibration of the carbonyl group, while the peak at 759 cm^−1^ corresponds to the stretching vibration of the aliphatic C-Cl bond. Furthermore, the absorption peak at 1343 cm^−1^ is assigned to the C-N stretching vibration.

In the blank PVDF and FR-2 films, the characteristic peaks observed at 1175 cm^−1^ and 1176 cm^−1^ are assigned to the stretching vibration of the C-F group in fluorocarbons [62], while the peaks at 1412 cm^−1^ and 1409 cm^−1^ correspond to the bending vibrations of C=C bonds. Notably, after loading the fluorescent probe FAM@RHB into the PVDF fibers, the characteristic peak at 769 cm^−1^ is attributed to the stretching vibration of the C-Cl bond in RHB. Additionally, the presence of the characteristic peak at 607 cm^−1^ in FR-2 is associated with the in-plane bending vibration of the keto structure (C-C=O) in 5(6)-FAM. This observation is consistent with the UV–Vis spectrum of 5(6)-FAM discussed in Section 3.3. Importantly, no new characteristic peaks emerged during the loading process, indicating that no new substances were formed and the original structures of the components remained intact.

### 3.9. Storage Stability of the Indicator Film

Fluorescent color stability plays a critical role in determining the indication accuracy of the indicator film. Figure 9 illustrates the fluorescence color changes in the indicator films after 14 days of storage at 4 °C. Generally, when Δ*E* is less than two, no significant color change is discernible to the naked eye [63,64]. In contrast, the FR-1 and FR-2 indicator films exhibited superior stability, with Δ*E* values of 1.16 and 1.24 after 8 days of storage, respectively. After 14 days of storage, their Δ*E* values increased only slightly to 1.71 and 2.05, respectively, rendering the color changes nearly invisible to the naked eye. On the other hand, the color differences of the FR-3, FR-4, and FR-5 indicator films were more pronounced, with Δ*E* values of 2.78, 3.58, and 4.89, respectively. This trend can be attributed to the gradual decrease in RHB content and the corresponding increase in 5(6)-FAM content from FR-1 to FR-5, as RHB demonstrated better photostability than 5(6)-FAM. In this study, all the films showed Δ*E* values lower than two within 7 days, implying that these films could be used for foods with a shelf life below 7 days.

### 3.10. Fluorescent Color Response of Indicator Films to TMA

As illustrated in Figure 10A, as the TMA concentration increased, the fluorescence colors of the five indicator films gradually shifted from yellow to yellow–green and finally to green, albeit with varying intensities. Specifically, the FR-1 indicator film initially exhibited an orange–yellow color in the absence of TMA. While the TMA concentration was elevated from 2.86 μM to 142.86 μM, the fluorescence color transitioned from orange–yellow to yellow–green, accompanied by an increase in the Δ*E* from 0 to 13.87. The FR-1 indicator film initially exhibited an orange–yellow color before reacting with TMA. During the increase in the TMA concentration from 2.86 μM to 142.86 μM, the film’s fluorescence color transitioned from orange–yellow to yellowish–green, accompanied by an increase in the Δ*E* from 0 to 13.87. However, the overall fluorescence color change remained relatively subtle. This phenomenon may be attributed to the low proportion of the fluorescent probe FAM in the FR-1 indicator film, resulting in its reduced responsiveness to TMA. Similarly, the FR-2 indicator film initially displayed an orange–yellow color in the absence of TMA. However, unlike the FR-1 film, the FR-2 film demonstrated significantly higher sensitivity to TMA. At a TMA concentration of 28.57 μM, the Δ*E* reached 16.19, and at 142.86 μM, the Δ*E* value increased to 21.01, with the fluorescence color transitioning from orange to green. In contrast, the FR-3 indicator film exhibited an initial fluorescence color of yellow, which shifted to green at the endpoint of the TMA reaction, corresponding to a Δ*E* value of 12.79.

The FR-4 and FR-5 indicator films initially exhibited a yellowish–green color, which transitioned to green at the reaction endpoint. At a TMA concentration of 142.86 μM, their Δ*E* values were 10.56 and 11.8, respectively. This can be explained by the higher content of FAM fluorescent probes in the FR-4 and FR-5 films, which required a greater amount of TMA to trigger a response. As a result, these films demonstrated comparatively poorer responsiveness than the FR-1 and FR-2 indicator films.

Additionally, the linear relationship between the fluorescence color change of the indicator films and the TMA concentration (ranging from 0 to 28.57 μM), as well as the sensitivity of the indicator films to TMA, were investigated. As shown in Figure 10B–F, there was a good linear relation between Δ*E* and TMA concentration in the range of 0–28.57 μM, with *R*^2^ values not less than 0.94. Based on calculations, the *LOD* values of the FR-1, FR-2, FR-3, FR-4, and FR-5 indicator films were determined to be 0.72, 0.54, 0.90, 1.06, and 1.10 μM, respectively. These results indicated that the FR-2 indicator film presented the best sensitivity to TMA.

Compared with other indicator films in Table 1, the fluorescent sensor developed in this study exhibited a lower *LOD* value, namely better gas sensitivity, making it suitable for practical application.

### 3.11. Application of Indicator Film in Livestock Meat Freshness Monitoring

The FR-2 indicator film exhibited excellent hydrophobicity, high stability, and strong sensitivity to TMA, making it an ideal candidate for monitoring the freshness of livestock meat. As shown in Figure 11A, the fluorescence color of the FR-2 film transitioned significantly from orange to green as the storage time increased and the pork and beef gradually spoiled. Furthermore, as illustrated in Figure 11B, by the third day of pork storage, the *TVB-N* content increased from 9.34 mg/100 g to 14.69 mg/100 g. According to the national standard [74], the maximum *TVB-N* value for pork is 15 mg/100 g, indicating that the pork on the third day is still safe to consume. However, by day 3.2, the *TVB-N* reached the critical threshold of 15 mg/100 g. At this point, the Δ*E* of the FR-2 indicator film was 7.4, and it showed a shift in fluorescence color from orange–yellow to yellow–green, signaling that the pork was no longer suitable for consumption. By the sixth day, the *TVB-N* value of the pork had increased to 25.88 mg/100 g, far exceeding the national standard, indicating serious spoilage. Concurrently, the fluorescence color of the indicator film turned green, with a Δ*E* value of 15.75. A similar trend was observed for beef, which reached the national standard limit of 15 mg/100 g on day 4.3. At this stage, the Δ*E* value of the indicator film was 9.2, and its fluorescence color transitioned from orange–yellow to green. The relationships between the Δ*E* values of the FR-2 film and the *TVB-N* values of beef and pork were shown in Figure 11D and Figure 11E, respectively. These results demonstrated that the FR-2 indicator film could effectively provide an early warning of pork and beef spoilage.

## 4. Conclusions

In this study, a composite ratiometric fluorescent probe was developed using 5(6)-FAM and RHB, which exhibited dual emission peaks at 516 nm and 580 nm under 490 nm excitation. SEM and laser confocal microscopy confirmed the uniform distribution of the fluorescent probes within the hydrophobic electrospun indicator film. The optimal mass ratio of 5(6)-FAM to RHB was determined to be 2:8 to develop the fluorescent film, namely the FR-2 film. The FR-2 film demonstrated superior sensitivity and stability, with an *LOD* value of 0.54 μM. The film maintained high stability after 14 days of storage at 4 °C, with a minimal change in the Δ*E* of only 2.05. The FR-2 indicator film used to monitor the freshness of pork and beef stored at 4 °C exhibited a distinct color transition from orange–yellow to green.

These findings highlight the potential of the FR-2 indicator film as a reliable tool for assessing the freshness of livestock meat in practical applications. The indicator film can be directly integrated into the headspace of existing meat packaging lids without direct food contact, responding to volatile compounds released during storage via visible color changes to indicate freshness levels. Unlike traditional indicators, this film requires UV light irradiation for color change observation, enabling freshness grading: consumers can use the included mini UV flashlight for detection, while producers may install fixed UV lamps on production lines for automated monitoring. PVDF offers exceptional chemical inertness, is non-toxic, is water-insoluble, and is free from F^−^ or small-molecule organic substance release. Listed by the U.S. FDA as a food-contact material, it is widely applicable in food packaging and beverage pipelines. However, Rhodamine B—a potentially toxic chemical—requires strict safety protocols for industrial and research use. Notably, the indicator film’s non-direct contact with meat products minimizes leakage risks. Future studies may explore whether the 5(6)-FAM to RHB leach from the films to the packaging environment or to the surface of meats.

## Figures and Tables

**Figure 1 foods-14-02200-f001:**
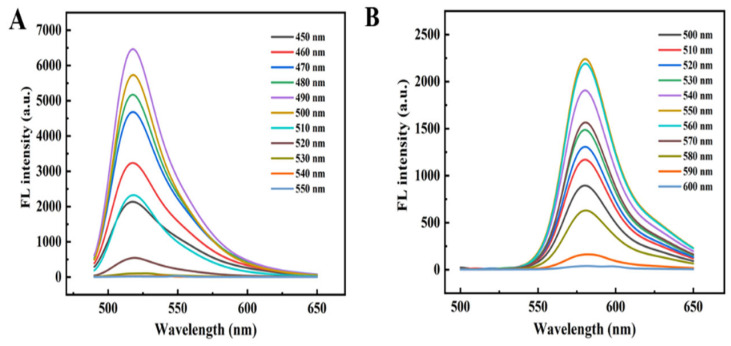
Fluorescence spectra of 5(6)-FAM (**A**) and RHB (**B**) at different excitation wavelengths.

**Figure 2 foods-14-02200-f002:**
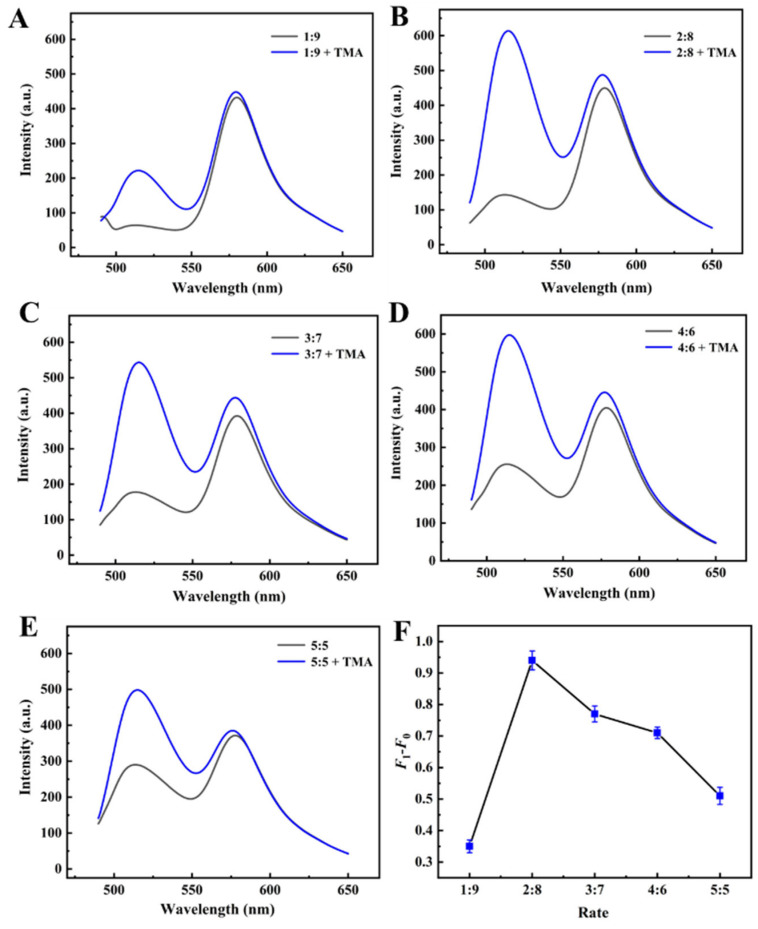
Fluorescence spectra (excitation wavelength of 490 nm) of the FAM@RHB probe solution at different volume ratios of FAM–RHB, before and after reacting with TMA (**A**–**E**); the *F*_1_*-F*_0_ of different ratios of fluorescent probes (**F**): *F*_1_ is *F*_516nm_/*F*_580nm_ after the reaction, and *F*_0_ is *F*_516nm_/*F*_580nm_ before the reaction.

**Figure 3 foods-14-02200-f003:**
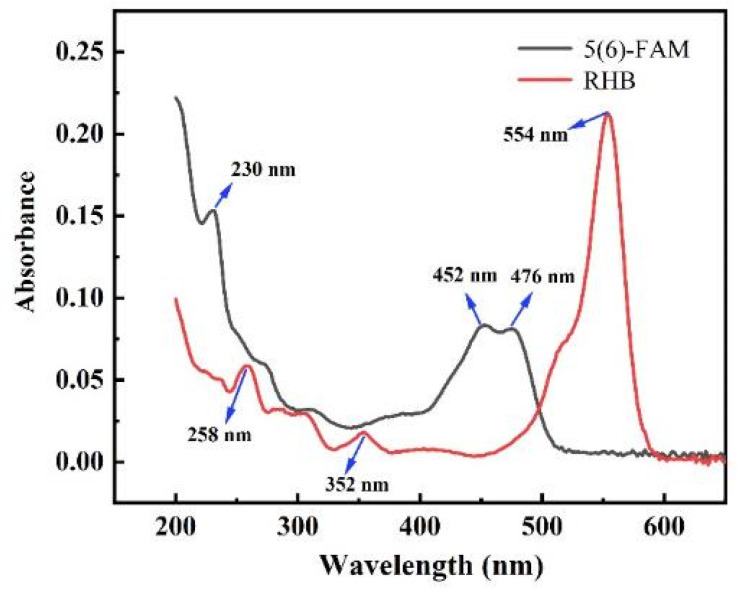
The UV–Vis absorption spectra of RHB and 5(6)-FAM.

**Figure 4 foods-14-02200-f004:**
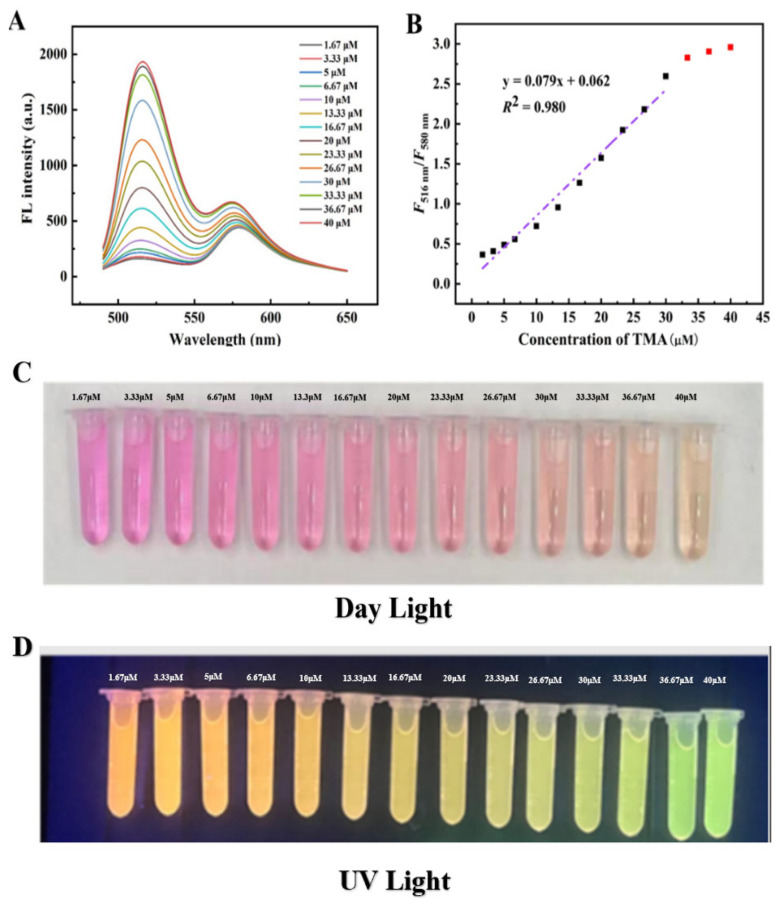
Fluorescence spectra of the FAM@RHB fluorescent probe (2:8) reacting with different concentrations of TMA under 490 nm excitation (**A**); linear relationship of the FAM@RHB ratiometric fluorescence intensity (*F*_516nm_/*F*_580nm_) versus TMA concentration (**B**); the color changes in the FAM@RHB fluorescent probe under daylight (**C**) and UV light (**D**) at different TMA concentrations.

**Figure 5 foods-14-02200-f005:**
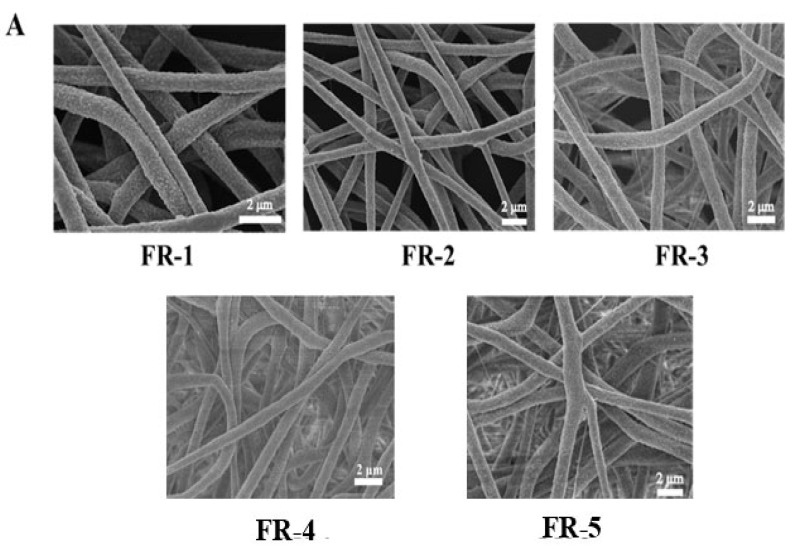
SEM images of FR-1, FR-2, FR-3, FR-4, and FR-5 films (**A**); fiber diameters of FR-1, FR-2, FR-3, FR-4, and FR-5 films (**B**).

**Figure 6 foods-14-02200-f006:**
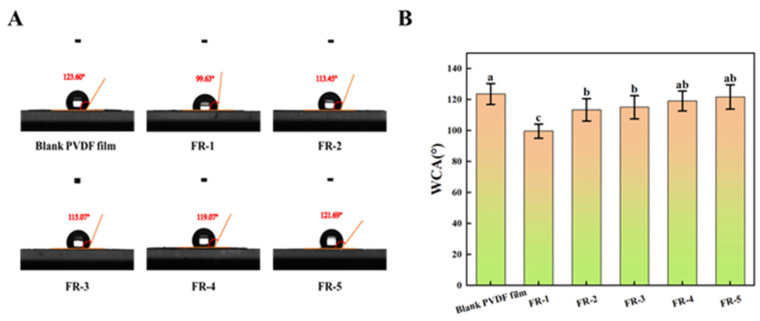
The WCA images (**A**) and significance analysis (**B**) of the pure PVDF film, FR-1 film, FR-2 film, FR-3 film, FR-4 film, and FR-5 film. Different letters in the figure indicate statistically significant differences (*p* < 0.05).

**Figure 7 foods-14-02200-f007:**
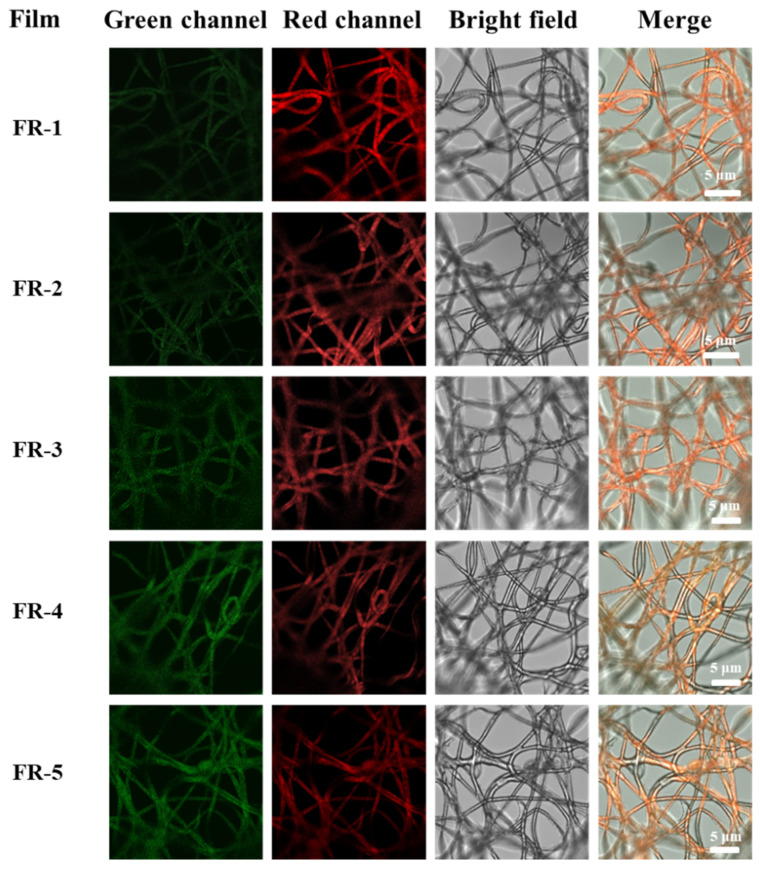
Confocal fluorescence images of five different films.

**Figure 8 foods-14-02200-f008:**
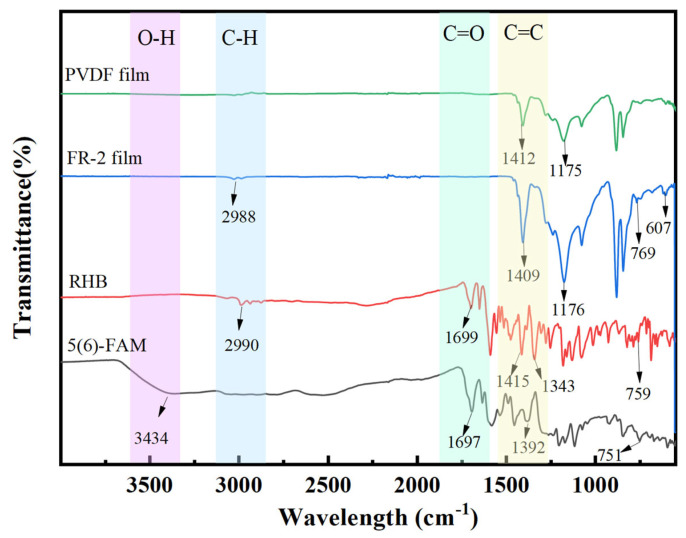
FTIR spectra of the FR-2 film.

**Figure 9 foods-14-02200-f009:**
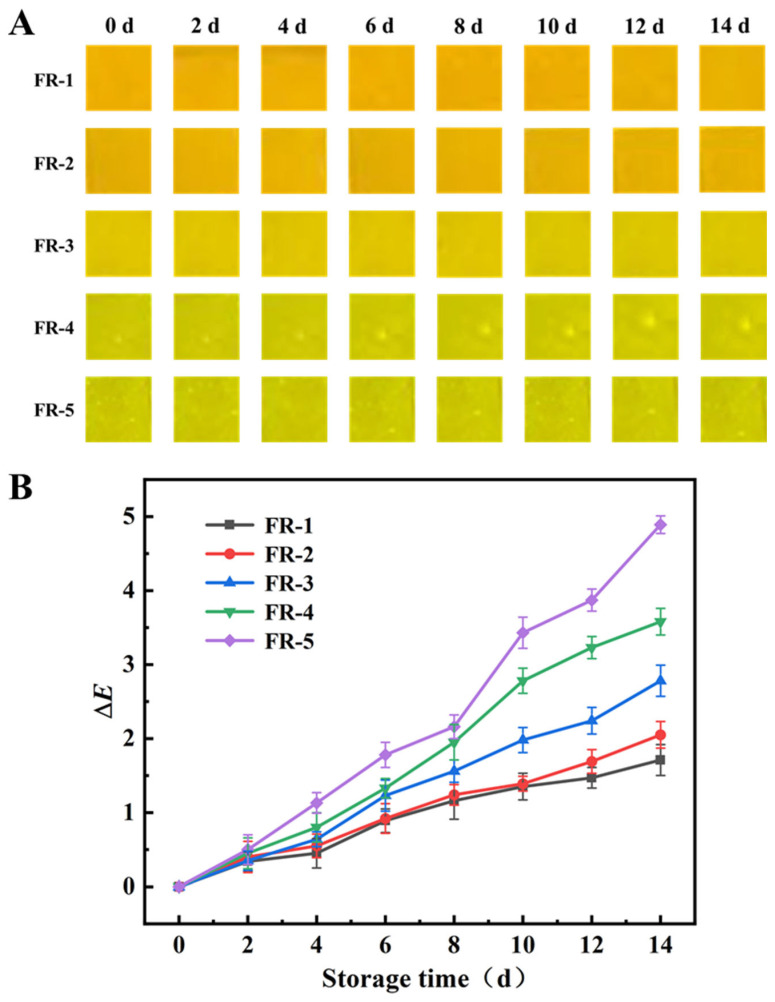
The fluorescent color changes (**A**) and color difference value Δ*E* change (**B**) in the films stored at 4 °C for 14 days.

**Figure 10 foods-14-02200-f010:**
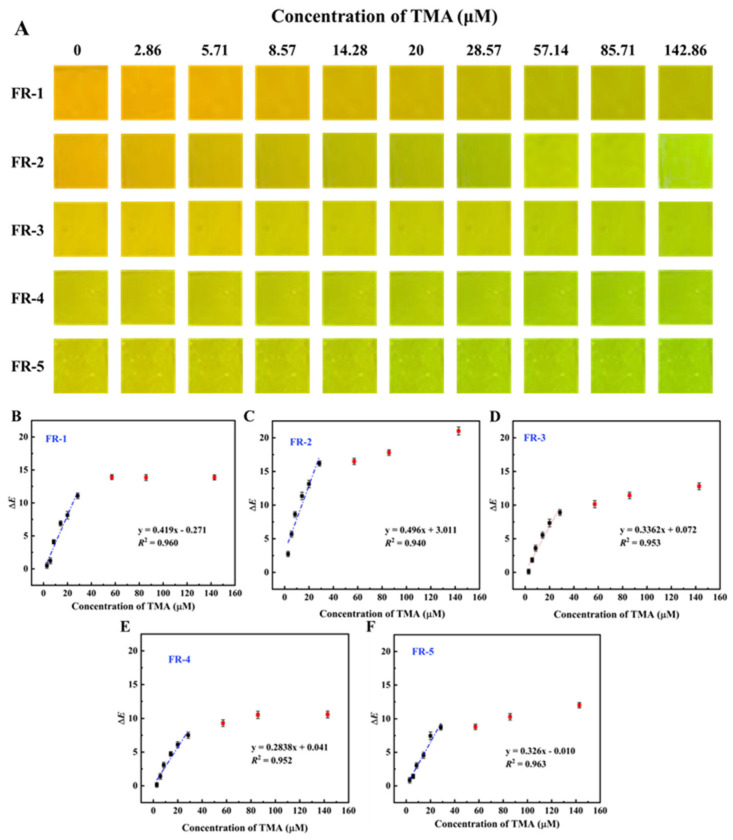
The colors of the FR-1, FR-2, FR-3, FR-4, and FR-5 films in response to TMA at different concentrations (**A**); the plot of the Δ*E* values of the FR-1, FR-2, FR-3, FR-4, and FR-5 films versus the concentration of TMA (**B**–**F**).

**Figure 11 foods-14-02200-f011:**
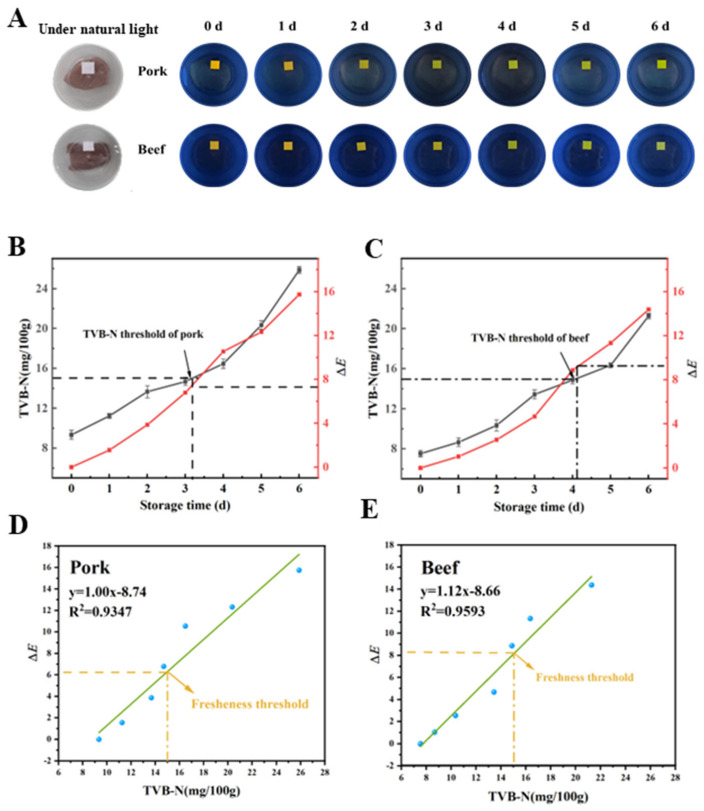
The photos of pork and beef packaging containing a FR-2 film under 4 °C under visible light and UV light (**A**). The relation between Δ*E* values of the FR-2 film and *TVB-N* values of pork (**B**) and beef (**C**). Correlation analysis of Δ*E* values of the FR-2 film with *TVB-N* values of pork (**D**) and beef (**E**).

**Table 1 foods-14-02200-t001:** A comparison of previously reported works and this work.

No.	Probe	Visible Light/Fluorescence	Analytes	*LOD*	Linear Range	Liquids or Solid Sensors for Use	Food Type	Ref.
1	βGlu-FITC	Fluorescence	Ammonia	——	——	PVA film	Pork	[14]
2	HPTS/TPB	Fluorescence	Ammonia	——	——	PVA film	Fish	[65]
3	BTCP-Ac@Eu-MOF	Fluorescence	Ammonia	0.68 ppm	——	Bacterial cellulose membrane	Chicken meat	[66]
4	FITC- Eu(Phen)_2_	Fluorescence	Ammonia	1.83 ppm	——	LCNF film	Shrimp	[67]
5	Tb^3+^ and Eu^3+^ co-doped silica	Fluorescence	Amine	——	——	Filter paper	Fish	[68]
6	ANs@MIL-2	Visible light	Ammonia	——	——	Pectin film	Chicken breasts	[69]
7	Cu-MOF	Visible light	Ammonia	7.31 μM	——	Agar film	Fish	[70]
8	FITC/PpIX	Fluorescence	Ammonia	5.0–2.5 × 10^4^ ppm	——	CA film	Shrimp	[16]
9	CeNPs	Visible light	HX	35 μM	6.2–200 μM	Gelatin film	Shrimp	[71]
10	HYM	Fluorescence	Putrescine	4.23 μM	1.259–5.428 μM	PVA film	Fish	[72]
11	N-GQD	Fluorescence	Ammonia	0.63 ppm	20–500 ppm	PVA film	Fish	[73]
12	5(6)-FAM/RHB	Fluorescence	Trimethylamine	0.54 μM	——	Electrospun films	Beef and pork	This work

## Data Availability

The data presented in this study are available upon request from the corresponding author. The data are not publicly available due to product development privacy.

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
