# Peer review of "A Hydrophobic Ratiometric Fluorescent Indicator Film Using Electrospinning for Visual Monitoring of Meat Freshness"

_foods, 2025, doi:10.3390/foods14132200_

Round 1

Reviewer 1 Report

Comments and Suggestions for Authors

Expand throughout the literature review so as to include a greater number of comparisons with alternative freshness indicators. The methodology that was used is thorough, though some details must be included in it. Film thickness as well as consistency across batches with the sample size used for meat freshness tests remain such details. Some sections could be condensed, but results seem clear. ∆E values as well as spoilage thresholds should be linked together. The link must be strengthened now. Explore the process of detecting spoilage compounds that are other than TMA. Figures get good preparation. You can include images showing what is the packaging setup and meat comparison both with and also without indicator film if that is what you want. Briefly discuss what challenges impede practical application like integration with packaging systems and UV light needs. External references give equilibrium yet self-citations stay chiefly relevant, though they might shrink.

The study is very well executed and is scientifically sound, but needs just minor revisions.

Comments on the Quality of English Language

The manuscript would gain from professional English editing for better grammar, flow, and clarity.

Author Response

Comment 1: Expand throughout the literature review so as to include a greater number of comparisons with alternative freshness indicators.

Response: Thank you for your suggestions. The relevant content regarding freshness indicator labels and sensors has been add in the manuscript. (Page 2, Line 39-48)

Comment 2: The methodology that was used is thorough, though some details must be included in it. Film thickness as well as consistency across batches with the sample size used for meat freshness tests remain such details.

Response: Thank you for your suggestions. We have supplement the following details: “For the same batch of purchased meat, part was used for application testing, while the other part was utilized to determine total volatile basic nitrogen (TVB-N). Both beef (loin) and pork (loin) were sampled from the longissimus dorsi muscle. Visible fat and connective tissues were excised, and sterile stainless-steel molds were employed to cut the meat into 60±1 g cuboids, ensuring uniform muscle fiber orientation. The thickness of the applied FR-2 film was approximately 126±2 μm” (Page 6, Line 223-229)

Comment 3: Some sections could be condensed, but results seem clear. ∆E values as well as spoilage thresholds should be linked together. The link must be strengthened now. Explore the process of detecting spoilage compounds that are other than TMA. Figures get good preparation. You can include images showing what is the packaging setup and meat comparison both with and also without indicator film if that is what you want.

Response: Thank you for your suggestions. The following content has been incorporated into the manuscript. “The relationships between the ∆E values of the FR-2 film and the TVB-N values of beef and pork were shown in Figures 11.D and E, respectively. These results demonstrated that the FR-2 indicator film could effectively provide early warning of pork and beef spoilage.”(Page 18-19, Line 490-493) .Additionally, we have included new application photos showing the FR-2 film indicating meat product freshness under visible light. (Figure 11 A)

Comment 4: Briefly discuss what challenges impede practical application like integration with packaging systems and UV light needs. External references give equilibrium yet self-citations stay chiefly relevant, though they might shrink. The study is very well executed and is scientifically sound, but needs just minor revisions.

Response: Thank you for your suggestions. The following content has been incorporated into the manuscript. “The indicator film can be directly integrated into the headspace of existing meat packaging lids without direct food contact. It responds to volatile compounds released during storage by undergoing visible color changes to indicate freshness levels.”(Page 19-20, Line 512-515)

“Unlike traditional indicators, this film requires UV light irradiation for color change observation, enabling freshness grading: consumers can use the included mini UV flashlight for detection, while producers may install fixed UV lamps on production lines for automated monitoring” (Page 20, Line 515-518)

Reviewer 2 Report

Comments and Suggestions for Authors

The presented study is well-structured and falls within the scope of food freshness monitoring using ratiometric fluorescent sensors. The study can be improved after addressing the following important comments:

1- The identity of PVDF is to be confirmed by either FTIR and/or EDX analysis, or other suitable techniques. This should be done for the neat and the modified electrospun PVDF nanofibers

2- In the introduction, discuss other studies used electrospun PVDF fibers for fabricating similar sensors. Examples from recent literature is: 

Song, Wenjun, et al. "A ratiometric fluorescence amine sensor based on carbon quantum dot-loaded electrospun polyvinylidene fluoride film for visual monitoring of food freshness." Food Chemistry 434 (2024): 137423.

3- Compare other types of the fluorescent materials used and show the novelty of your proposed method in the last part of the introduction.

4. How can consumers use the sensor under UV? How can this limitation be tackled? provide some literature support for your answer. discuss this issue in the text related to food testing.

5- What about the toxicity of the dye Rhodamine B and the PVDF fluorinated polymers? Please discuss this in the future directions/conclusions. 

6- It would be appreciated if the authors could explain the high hydrophobicity of the electrospun PVDF nanofibers (124 degrees or above).

7- It is required to highlight the importance of hydrophobicity in the food freshness indicators with recent literature reports using different approaches to reflect the novelty of this work.

8- In conclusions, summerize  the results for the food freshness test as the delta E values and color change. Also, add a statement about the possibility of using this kind of sensor for other food monitoring applications. ‏

Author Response

All comments were addressed in a Word document.

The presented study is well-structured and falls within the scope of food freshness monitoring using ratiometric fluorescent sensors. The study can be improved after addressing the following important comments:

Comment 1: The identity of PVDF is to be confirmed by either FTIR and/or EDX analysis, or other suitable techniques. This should be done for the neat and the modified electrospun PVDF nanofibers.

Response: Thank you for your suggestions. The infrared spectrum of the blank PVDF film has been shown in Figure 8.

Comment 2:  In the introduction, discuss other studies used electrospun PVDF fibers for fabricating similar sensors. Examples from recent literature is: Song, Wenjun, et al. "A ratiometric fluorescence amine sensor based on carbon quantum dot-loaded electrospun polyvinylidene fluoride film for visual monitoring of food freshness." Food Chemistry 434 (2024): 137423.

Response: Thank you for your suggestions. The relevant content has been supplemented in the manuscript.

“Our recent study successfully developed an ratiometric fluorescent sensor using synthesized carbon dots as the probe [44]. In comparsion, this study employed both Rhodamine B and 5(6)-FAM as probes due to their commercial availability, which renders the fluorescent sensor more conducive to commercialization..”(Page 3, Line 85-88)

Comment 3:  Compare other types of the fluorescent materials used and show the novelty of your proposed method in the last part of the introduction.

Response: Thank you for your suggestions. The comparison with indicator films in other articles is all included in Table 1. “Compared with other indicator films in Table 1, the fluorescent sensor developed in this study exhibited lower LOD value, namely better gas sensitivity, making it for practical application.”(Page 16, Line 462-464)

Comment 4: How can consumers use the sensor under UV? How can this limitation be tackled? provide some literature support for your answer. discuss this issue in the text related to food testing.

Response: Thank you for your suggestions. The further development of freshness sensors can be carried out in the following aspects.

Consumer Instructions: Consumers can use a standalone UV flashlight to irradiate the indicator film and observe the color change of the film. Mobile APP integration: Develop a supporting mobile application to capture images of the indicator film under UV excitation via the phone camera. Leverage image recognition algorithms to analyze color discrepancies (such as variations in RGB values), and directly present freshness grades (e.g., "fresh", "critical", "spoiled") [1].

Instructions for Production Use: The UV lamp can be fixedly installed on the production line, and computer vision analysis can be employed to determine the product freshness grade based on the color of the indicator film, thereby enabling quality control of the products.

Comment 5: What about the toxicity of the dye Rhodamine B and the PVDF fluorinated polymers? Please discuss this in the future directions/conclusions.

Response: Thank you for your suggestions. It has been fully supplemented in the manuscript. “PVDF offers exceptional chemical inertness, being non-toxic, water-insoluble, and free from F⁻ or small-molecule organic substance release. Listed by the U.S. FDA as a food-contact material, it is widely applicable in food packaging and beverage pipelines. However, Rhodamine B—a potentially toxic chemical—requires strict safety protocols for industrial and research use. Notably, the indicator film's non-direct contact with meat products minimizes leakage risks. Future studies may explore whether the 5(6)-FAM to RHB leach from the films to the packaging environment or to the surface of meats.” (Page 20, Line 518-525)

Comment 6:  It would be appreciated if the authors could explain the high hydrophobicity of the electrospun PVDF nanofibers (124 degrees or above).

Response: Thank you for your suggestions. PVDF's strong hydrophobicity stems from its unique molecular and structural characteristics. The fluoropolymer's C-F bonds create a low-energy surface, while electrospun nanofiber roughness enhances this effect through air-trapping microstructures. Together, these features produce exceptional water repellency with contact angles well above 90°.

Comment 7: It is required to highlight the importance of hydrophobicity in the food freshness indicators with recent literature reports using different approaches to reflect the novelty of this work.

Response: Thank you for your suggestions. During the storage of meat products, the humidity in the packaging will increase. When the humidity in the food packaging is high, the hydrophilic indicator film is prone to dissolve in water, thus damaging its integrity [2]. In addition, the hydrophilicity of the indicator film may cause the dye to diffuse from the solid matrix to the film surface, potentially leading to leaching and leakage of the dye. This can result in a decrease in dye concentration and uneven distribution within the film, thereby reducing its indication accuracy in high-humidity environments [3,4].

Comment 8: In conclusions, summerize the results for the food freshness test as the delta E values and color change. Also, add a statement about the possibility of using this kind of sensor for other food monitoring applications. ‏

Response: Thank you for your suggestions. The ratio fluorescent indicator film developed in this study can be applied to the freshness detection of seafood in the future. During the spoilage of fish and shellfish, characteristic gases such as trimethylamine (TMA) will be released. Meanwhile, sensors prepared by combining hydrophobic polymers like PVDF can effectively resist the interference of high-humidity environments of aquatic products on the film performance.

  1. K. P, C.; T. P, V.; Nagarajan, P. Smartphone application-based colorimetric fish freshness monitoring using an indicator prepared by rub-coating of red cabbage on paper substrates. Colloids and Surfaces A: Physicochemical and Engineering Aspects 2023, 679, 132553, doi:https://doi.org/10.1016/j.colsurfa.2023.132553.
  2. Sabu Mathew, S.; Jaiswal, A.K.; Jaiswal, S. A comprehensive review on hydrophobic modification of biopolymer composites for food packaging applications. Food Packaging and Shelf Life 2025, 48, 101464, doi:https://doi.org/10.1016/j.fpsl.2025.101464.
  3. Kalita, P.; Bora, N.S.; Gogoi, B.; Goswami, A.; Pachuau, L.; Das, P.J.; Baishya, D.; Roy, S. Improving the hydrophobic nature of biopolymer based edible packaging film: A review. Food Chemistry 2025, 479, 143793, doi:https://doi.org/10.1016/j.foodchem.2025.143793.
  4. Wang, Y.; Jiang, L.; Liu, T.; Qin, R.; Xue, J.; Wang, L.; Ren, L. Duck feather inspired hydrophobic carboxymethyl cellulose/gelatin film for intelligent and active food packaging. International Journal of Biological Macromolecules 2025, 301, 140424, doi:https://doi.org/10.1016/j.ijbiomac.2025.140424.

Reviewer 3 Report

Comments and Suggestions for Authors

The manuscript is written based on the development and study of intelligent packaging for the indicating and monitoring the meat freshness during refrigerating. 

In the method part the well known and worldwide used methods are described but their literature origin and usage is not cited.

The 3rd part should be entitled like "Results and Discussion"

The manuscript is poorly discuss the results with the international literature. The differences, advantages or disadvantages between that packaging and other similar intelligent packaging in the literature is not presented. Some of the literature resources are missing from the list (52-56).

Comments on the Quality of English Language

There is a few spelling mistakes, which should be corrected carefully.

Author Response

All comments were addressed in a Word document.

The manuscript is written based on the development and study of intelligent packaging for the indicating and monitoring the meat freshness during refrigerating.

Comment 1: In the method part the well-known and worldwide used methods are described but their literature origin and usage is not cited.

Response: Thank you for your suggestions. The following content has been incorporated into the manuscript. References 51 have been inserted at line 192 for the reference method of color determination of indicator films, References 46 at line 143 for the method parameters of electrospinning, References 8 at line 131 for the relevant literature on the preparation of PVDF solutions, and References 45 at line 123 for the relevant literature on FAM fluorescence spectra. (Page 3-4)

Comment 2: The 3rd part should be entitled like "Results and Discussion"

Response: Thank you for your suggestions. The title of the third part has been changed to "Results and Discussion". (Page 6, Line 248)

Comment 3: The manuscript is poorly discusses the results with the international literature. The differences, advantages or disadvantages between that packaging and other similar intelligent packaging in the literature is not presented.

Response: Thank you for your suggestions. The comparison between this study and other intelligent packaging has been added to Table 1. (Page 16, Line 462-464)

Comment 4: Some of the literature resources are missing from the list (52-56).

Response: Thank you for your suggestions. References in the article have been supplemented.

Round 2

Reviewer 2 Report

Comments and Suggestions for Authors

All comments , suggestions, and required revision have been adequetly addressed. 

Author Response

We sincerely appreciate your valuable suggestions.

Reviewer 3 Report

Comments and Suggestions for Authors

The manuscript has been corrected and extended.

Author Response

We sincerely thank you for your valuable suggestions.